# Preparation of Chitosan/Corn Starch/Cinnamaldehyde Films for Strawberry Preservation

**DOI:** 10.3390/foods8090423

**Published:** 2019-09-19

**Authors:** Yue Wang, Rui Li, Rui Lu, Jie Xu, Ke Hu, Yaowen Liu

**Affiliations:** 1College of Food Science, Sichuan Agricultural University, Yaan 625014, China; yueerjiejie520@163.com (Y.W.); 18980342538@163.com (R.L.); lr548624lr@163.com (R.L.); 18398264931@163.com (J.X.); yl038024097@126.com (K.H.); 2School of Materials Science and Engineering, Southwest Jiaotong University, Chengdu 610031, China; 3California NanoSystems Institute, University of California, Los Angeles, CA 90095, USA

**Keywords:** chitosan, corn starch, cinnamaldehyde

## Abstract

In this study, the casting method was used to make chitosan (CS)/corn starch/cinnamaldehyde film, and the preservation performance of the film was examined. The results showed that the tensile strength of the film can reach to 31.24 ± 0.22 MPa when the mass ratios of CS, corn starch, and glycerin were 2.5%, 7%, and 0.5% respectively. The addition of cinnamaldehyde made the films have great inhibitory effect on *Botrytis cinerea, Rhizopus*, and *Escherichia coli.* In particular, the film had a significant fresh-keeping effect on strawberries, which reduced the loss of nutritional value, when aiming at soluble solids, titratable acid value, weight loss rate, and other indexes of strawberries. Thus, the films can slow down the physiological changes of strawberries and extend their shelf life to 11 days. Therefore, this work demonstrates the noteworthy potential of these novel films, incorporating natural antimicrobial compounds as innovative solutions to be used in active food packaging to extend the shelf-life of food products.

## 1. Introduction

Environmentally-friendly preservative films usually possess excellent mechanical and barrier properties, but poor antibacterial properties. Therefore, it has become a research hot-spot to strengthen the antibacterial properties of preservative films; prevent the food and films themselves from microbial pollution; and develop an environmentally-friendly antibacterial film which is applied safely, with a broad-spectrum, and high efficiency. An environmentally-friendly antibacterial preservative film is a kind of film which is prepared by adding appropriate amount of antibacterial agent to the prepared film-forming liquid, and achieves the purposes of antibiosis and preservation through the sustained release of the antibacterial agent, and the shock absorption, protection, and isolation of the film through molecular crosslinking. Antibacterial agents are generally divided into three categories: inorganic antibacterial agents, organic antibacterial agents, and natural antibacterial agents [1]. With people more and more inclined to natural, organic, and green food processing, how to apply natural antibacterial agents in the field of food processing effectively has gradually become the focus of research. The natural antibacterial agents which can be seen in normal life are antibiotics (such as *Lactococcus peptide*), lysozymes, plant essential-oils, etc [2,3]. Plant essential-oils are also known as volatile oils, which are mainly extracted from unique aromatic substances in plants, such as flowers, leaves, roots, bark, fruit, seeds, resins, and so on [4,5]. Plant essential oils are safe, non-toxic, and have certain antibacterial properties. They are green and healthy, natural preservative agents. Du and et al. [6] studied the physicochemical and antibacterial properties by adding cinnamon’s essential oil, sweet pepper’s essential oil and clove bud’s essential oil to CS film respectively. The results showed that cinnamon’s essential oil had the least effect on the physiochemical properties of CS film. At the same time, the antimicrobial effect of cinnamon’s essential oil was greater than clove’s essential oil, while clove’s essential oil’s effects were greater than those of sweet pepper’s essential oil. Avila-Sosa et al. [7] added lemon citronella’s essential oil, oregano’s essential oil, and cinnamon’s essential oil to amaranth, CS, and starch edible film; then, CS film showed better antifungal effects than the others. Among the many essential oils, cinnamaldehyde shows particularly good performance in bacteriostasis and safety, and has a strong aromatic smell, which is often used as a blending, preservative, and anti-mildew agent in food. It has been listed as a food additive and edible flavor in China [8]. The FDA of the United States also regards it as a food-grade chemical [9]. Cinnamaldehyde, one of the major constituents of cinnamon’s bark oil, possess high antibacterial, antifungal, antiinflammatory, and antioxidant activity. Several studies have demonstrated that cinnamaldehyde could be potentially used as an effective antimicrobial agent in active packaging. As a result, some researchers used cinnamaldehyde in food preservation and antibacterial films [10,11]. It is reported that cinnamaldehyde has an obvious bacteriostatic effect on *Bacillus subtilis, Escherichia coil, Staphylococcus aureus,* and *Salmonella*, and the antibacterial effect is enhanced when the concentration of essential oil increases [12]. Hu et al. [13] also believed that reactive oxygen species (ROS) produced by cinnamaldehyde benzylation can lead to cell death, programmed death, and so on; and even cancer cell death in fungal metabolic activities. Hassani et al. [14] found in their study that the hydrophilic group in cinnamaldehyde can destroy the outer film of the cell and enhance the permeability of the cell film, which makes the cell adenosine triphosphate exudate, and leads to cell death. Ouattara et al. [15] added cinnamaldehyde or lauric acid to a chitosan matrix to prepare an antimicrobial film and applied it to regular cooked ham and pastrami. The bacteriostatic test showed that the film containing cinnamaldehyde had higher antibacterial activity. The successful preparation of the tapioca starch/cinnamaldehyde film was finished by de Souza et al. [16]; it demonstrated antibacterial activity against fungi commonly found in bread products. Balaguer et al. [17] used a gliadin film containing cinnamaldehyde in an active food packaging system for slicing bread and cheese, which proved that the film could increase the shelf life of sliced bread and cheese sauce. Higueras, et al. [18] studied the hydrolysis of imino bonds and the subsequent release of cinnamaldehyde after the films had been subjected to different combinations of temperature/time treatments and simulating food preservation methods, and evaluated the comprehensive performance of cinnamaldehyde-imine-chitosan film. Besides, the milk was used as an example to show that the smell of cinnamon does not cause any rejection by potential consumers. Those all illustrate the great potential of cinnamaldehyde for active food packaging. In addition, chitosan (CS) is a new kind of bio-material, which has been widely used in food preservation and packaging recently for its good biocompatibility and degradability. Bonilla et al. [19] used CS instead of wheat starch in the research on the effect of CS on the physical properties of wheat starch/glycerin film and found that the tensile strength and elastic modulus of the film could be improved by increasing the CS–starch ratio. It was worth mentioning that the film had good bactericidal activity, especially when the CS–starch ratio was 50%. At present, most of the natural active ingredients are added to the film-forming substrates to form antibacterial fresh-keeping films, and natural ingredients are slowly released in film-forming materials, which can inhibit the growth of microorganisms. Victor et al. [20] noticed that the prepared nano-clay/starch carvol/thymol bacteriostatic film can effectively inhibit the activity of gray mold and has a good bacteriostatic effect, while not affecting the quality parameters of a strawberry’s fruit (such as firmness, quality, appearance color, etc.).

The strawberry (*Fragaria’ ananassa Duch.strawberry*) is a kind of delicate fruit, whose water content is high, and it is easily perishable. Its shelf life is usually 1 or 2 days, which means it will become valueless in days. At present, physical and chemical methods are commonly used for strawberry preservation [21]. Although physical preservation methods, such as controlled atmosphere, low temperature, irradiation, and vacuum, etc. have certain effects on the strawberry preservation, they require large equipment, and the cost is too high. Besides, the chemical synthetic substances may have adverse effects on human health, and even cause cancer, teratogenicity, and mutagenesis. In addition, although the coating preservation method has a certain effect on the preservation of strawberries, the surface of the strawberry is covered with a layer of wax, which makes it difficult to apply the film evenly. As the result, the part of the fruit’s surface which cannot be covered will appear with white spots and rot, affecting the overall preservation effect.

At present, there is no research on the effect of chitosan/corn amylose/cinnamaldehyde on the preservation of the strawberry. Therefore, the purpose of this study is to study the effect of adding different ratios of cinnamaldehyde, on the material properties of CS/corn starch films, through a series of tests on mechanical performance, barrier properties, and bacteriostatic properties, etc. In addition, the strawberry preservation experiments were carried out using the film with the optimal cinnamaldehyde content, which will provide the foundation for the subsequent research.

## 2. Materials and Methods 

### 2.1. Materials

CS with the deacetylation degree of 95% and the molecular weight (Mw) of 2.8 × 10^5^ was purchased from Golden-Shell Pharmaceutical Co. Ltd. (Zhejiang, China), principally from crabs and shrimps. Cinnamic aldehyde was purchased from Guoguang Perfumery Factory (Ji’an, China). Corn starch was purchased from Chengdong Food Co., Ltd. (Chengdu, China). The glycerin (purity 99.97%, density 1.26 g/cm^3^) used in this study was supplied by Chengdu Kelong Chemical Reagent factory. All other chemical reagents used were of analytical grade and obtained from Chengdu Kelong Reagent Co. (Chengdu, China) unless otherwise indicated.

### 2.2. The Preparation Process and Performance Evaluation of CS/Corn Starch Films 

Firstly, 1% acetic acid solution (*w*/*w*) was prepared; then CS was added. A magnetic stirrer was stirring at 50 °C for 90 min, until the CS completely dissolved and formed different CS solutions (2, 2.5, and 3 wt%). The appropriate amount of corn starch was then placed in a water bath (Zhongxing Weiye Instrument Co., Ltd. Beijing, China) at 80 °C and stirred for 120 min to obtain a gelatinized corn starch solution (5, 6, and 7 wt%). Finally, the solution obtained in the above step was mixed at a ratio of 5:5, and stirred at 60 °C for 30 min in a water bath, then glycerin was added (0.5, 1, and 1.5 wt%), and vacuum ultrasonic was performed to obtain a CS/corn starch complex solution. The prepared chitosan/corn starch solution was cast in the mold (glass plate 30 × 40 cm), and then put it into the oven at 50 °C, and the baking time was 6–8 h. After that, the mold was taken out and cooled to a room temperature, and the film was peeled off, and placed at 25 °C, 50% relative humidity (RH) for storage. The performance of the film was evaluated by a scoring system with a water permeability of 40 score, a tensile strength of 30 score, and an elongation at break of 30 score. Calculation rules: score per item = item score ranking coefficient. Total score = ∑ each score. The ranking coefficient is decremented according to the ranking; e.g., for first place, the coefficient is 1.0; for second place, the coefficient is 0.8; and for third, the coefficient is 0.6. (The lower the water permeability coefficient is, the higher the ranking is; and the higher the tensile strength and elongation at break is, the higher the ranking is.)

### 2.3. Orthogonal Experimental Design

The most appropriate chitosan solution mass ratio, corn starch mass ratio, and glycerol mass ratio were selected for 3 factors and 3 levels of orthogonal experimental design. An L9 (33) orthogonal table was selected for experimental arrangements to study the influence of various factors on the preservation performance of the film.

In the mixed solution obtained under the above process conditions, cinnamaldehyde was added with 50% ethanol as a co-solvent, and the film was mechanically blended at room temperature for 20 min to form a film.

### 2.4. Film Color Difference 

Color coordinates, L* (lightness), a* (red-green), and b* (yellow-blue) were measured at room temperature. The instrument was calibrated with a white standard tile. Measurements were carried out in quintuplicate at random positions over the film surface. Average values for these five tests were calculated. Total color difference (△E) was calculated as follows: $$\Delta E=\sqrt{\Delta a^{2}+\Delta b^{2}+\Delta L^{2}}$$

### 2.5. Mechanical Properties 

Bubble-free and flat antibacterial films were selected, cut into dumbbell shapes by a die cutter, and mounted on a micro-control electronic universal testing machine (Super Technology Instrument Co., Ltd. Shanghai, China) to perform elongation at break (E) and tensile strength (TS, tensile strength) tests at room temperature. The original distance (L_0_) between the tops and bottoms of the tested-samples was 50 mm and the film material was stretched at 50 mm/min until the samples were fractured. The data was recorded by the computer during the test, and the E and TS values were read directly from the computer. Three sets of data were measured for each set of samples and the average was taken.

### 2.6. Water-Vapor Transmission Rate (WVP) 

A previously reported method was used, as a reference [22]. After sample fixation, silica gel was placed in a drying tower with 0% relative humidity at 25 °C. A cup with an aluminum foil was used as the control sample to estimate the loss through the seal. After the steady state reached, an analytical balance (0.0001 g) was used to weigh the cup every 2 h. The water-vapor permeability coefficient was calculated from the steady-state permeability slope over time in the weight loss curve. Weight loss was calculated by subtracting loss through the seal from the total weight loss. The fiber thickness was measured to obtain the water-vapor permeability. Three sets of data were measured for each sample and the average was taken.

### 2.7. The Antibacterial Effect of CS/Corn Starch/Cinnamaldehyde Films on Strawberry

The filter paper diffusion method was adopted according to the method of Shojaeealiabadi et al. with slight modifications [23]. The activated *Rhizopus*, *Botrytis cinerea* and *Escherichia coli* (10^−7^ CFU/mL) were diluted to 10^−6^ CFU/mL; 100 μL of the bacterial solution was uniformly coated on the inactivated plate agar. The bacteriostatic film with a diameter of 6 mm was sterilized on both sides by ultraviolet light for 2 h (1 h each on the front and back sides, and then placed flat on the coated plate, and each dish was sealed by a plastic wrap to prevent the cinnamaldehyde from evaporating. They were cultured at 25 °C for 72 h (*E. coli* was cultured at 37 °C for 48 h). A plate without any film was used as a blank, and a polypropylene (PP) film was used as a control, and each treatment used triplicates in parallel. Similarly, the film without cinnamaldehyde was set up to carry out a bacteriostatic experiment through the above method; each treatment used three replicates in parallel. The inhibition zone was measured by a vernier caliper. When the inhibition zone value was not significant, the low-concentration cinnamaldehyde was preferred for the following experiment.

### 2.8. Strawberry Preservation Experiments

Strawberry was selected for preservation experiment, and the optimal cinnamaldehyde concentration of CS/corn starch/cinnamaldehyde (1.6%) bacteriostatic film, ordinary PP film, and CS/corn starch film were selected as the experimental groups. At the same time, a sample without film was used as a control group named CK group. Strawberries were collected from Tianfu Lv Kang Farm in the Shuangliu District, Chengdu City, Sichuan Province. For every group, three strawberries were placed on acrylic plates and wrapped around with the films. All samples were stored at room temperature (20 ± 1 °C) with relative humidity: 65% ± 5%. The strawberries were measured every 2 days for a total of 7 measurements in a confined room. According to the size of the fruit decay areas, the degrees of decay were divided into 5 grades: Grade I, no decay; Grade II, the area of decay accounts for 25% of the fruit area; Grade III, the area of decay accounts for 25% to 50% of the fruit area; Grade IV, the area of decay accounts for 50% to 75% of the fruit area; Grade V, the decayed area accounts for 75% or more of the fruit area. An analytical balance (0.0001 g) (Haoyu Hengping Scientific Instrument Co., Ltd. Shanghai, China) was used to measure the weight of the strawberry to determine the weight loss rate. The initial percentage of weight loss is used to indicate the weight loss rate of the strawberry. Three strawberries in each group were weighed three times.
Weight loss rate (%) = (W_0_ − W_t_)/W_0_ × 100%

In the formula, W_0_ is the weight of fresh strawberries and W_t_ is the weight of stored strawberries.

A texture analyzer (Texture Technologies Corp., Scarsdale, NY, USA) equipped with a 50 kg load cell was used to measure the firmness of the strawberries at room temperature after wrapping them using the films and before treatment. A single strawberry was placed on a platform, and a flat stainless steel cylindrical probe of 2 mm in diameter punctured 6 mm into the fruit, at a speed of 1 mm/s [24]. After the firmness test, the strawberries were cut into small pieces and high-speed homogenization was performed for 2 min with a handheld homogenizer. The strawberry homogenate was divided into groups of 5 g, 5 g, and 10 g. For titratable acid analysis, 5 g of the strawberry homogenate was placed in a 50 mL volumetric flask filled with distilled water. The flask was left to stand for 30 min before filtration, and 20 mL of the filtrate was then titrated with 0.01 mol/L NaOH. The following formula was used to calculate the total titratable acidity of the diluted strawberry pulp.

TA = V_NAOH_ × 0.1 × 0.064 × V_0_/m × V_1_

In this formula, V_NaOH_ is the volume of NaOH spent for titration in mL, V_1_ is the molarity of the NaOH solution, 0.064 is the conversion factor for citric acid, V_0_ is the volume of the volumetric flask, and m is the filtrate volume. The results were expressed in citric acid equivalents (g citric acid/100 g dry weight). For analysis of total soluble solids (TSS), 5 g of strawberry homogenate was added to 10 mL distilled water and then filtered with filter paper. A handheld refractometer was used to measure the TSS. Triplicate measurement samples and average values were calculated by the standard curve of pure ascorbic acid. In brief, 100 mL of 50 g/L trichloroacetic acid was used for the extraction of 10 g of the strawberry homogenate. After 10 min, the extract was centrifuged, and the clarified supernatant was collected. The supernatant was added to a 1.0 mL test tube, and 1.0 mL of 50 g/L trichloroacetic acid, 1 mL of 5 g/L phenanthroline/ethanol solution, and 0.5 mL of 0.3 g/L FeCl_3_/ethanol solution were added to the test tube. Absorption spectrophotometry was used to measure the absorbance of the test sample at 534 nm.

### 2.9. Data Analysis

Data analysis was processed by Excel and SPSS stats software. The data results are expressed as averages ± standard deviations. Different letters were used to indicate a difference between the same indicator data (*p* < 0.05), and the same letter indicates no difference; the drawing was performed using Origin 8.1 software (Microcal Software Inc., Northampton, MA, USA).

## 3. Results

### 3.1. Orthogonal Experimental Results of CS/Corn Starch/Glycerin Films 

The orthogonal experimental design of three factors and three levels was adopted. L9 orthogonal table was selected to arrange the experiment, and the influences of various factors on the physical and chemical properties of the film were studied (Table 1). The WVTR of a film is mainly determined by the arrangement and crystallinity of the film’s polymer molecules. The high amylose corn starch film is dense and neat, forming hydrogen bonds within itself, so that water molecules are difficult to penetrate the film [25]. It can be seen from Figure 1, that the permeability coefficient of the film at different ratios presents an irregular distribution, of which the lowest, in the group 212 (CS (wt%):corn starch (wt%):glycerin (wt%) = 2:1:2) is 1.72 ± 0.03 × 10^−12^ g cm/(cm^2^ s Pa). This was because when CS was mixed with a high amylose corn starch solution to form a film, the amino group of CS and the hydroxyl groups of high amylose corn starch underwent physical and chemical cross-linking to form hydrogen bonds, which affected the crystallinity of the molecule and enhanced the water-resistance of the film. In addition, glycerin as a plasticizer can adjust the molecular order of the film to change its compactness.

Similarly, according to Figure 2, TS and E values of thin films at different ratios show different laws, with the highest TS value being 31.38 ± 0.74 MPa, and the highest E value was 47.96% ± 1.93%. In Table 2, according to the range analysis, it can be seen that the main factors affecting the orthogonal results were glycerin, CS, and corn starch, respectively, according to the range calculations. Glycerin, as a commonly used plasticizer, is known for its ability to regulate the level of film plasticity. The addition of glycerin can increase the distance between CS and corn starch molecules, weaken the cross-linking between them, counteract the hydrogen bonding and reduce the tensile strength of the films. At the same time, CS was similar to glycerin as a film-forming substrate with good hydrophilicity. Usually, the addition of CS will combine both advantages and form complementary advantages. According to Table 2, the optimum condition of orthogonal modularity was finally determined to be the 231 (experimental group 231) combination through range analysis. According to variance analysis in Table 3, it can be seen that factors A and C had a significant influence on the physical and chemical properties of the thin films, while factor B had no significant influence.

### 3.2. Film Color Difference of the Cs/Corn Starch/Cinnamaldehyde Films

According to the results of the orthogonal experiment (CS 2.5%, corn starch 7% and glycerin 0.5%), the CS/corn starch/cinnamaldehyde film was prepared by adding the cinnamaldehyde. The results of the film color difference of the film are shown in Table 4. In general, the film color should be close to colorless or the commonly used polymer film. It can be seen from Table 4 that the lightness of the film decreases with the increase of cinnamaldehyde content. The color of the film gradually changed to green and yellow. The film color did change significantly (*p* < 0.05), which was due to the yellowish green color of cinnamaldehyde. It can be judged that cinnamaldehyde was evenly distributed in the film. According to the change of ΔE value, the increase of cinnamaldehyde content obviously affected the color difference of film. The more the cinnamaldehyde content there was, the greater the ΔE content was. The color difference of 0.4% cinnamaldehyde was 300.77% of the experimental group 231′s color difference. The color difference of 6.4% cinnamaldehyde was 435.89% that of 0.4% cinnamaldehyde. Changes in the color characteristics of films enriched with volatile compounds could be explained by some alterations of the macromolecular structure which may have occurred when volatile compounds were added; however, more analyses are necessary to confirm this explanation [26]. 

### 3.3. Mechanical Properties of the CS/Corn Starch/Cinnamaldehyde Film

Figure 3 showed that the tensile strength and elongation at the breaking point of the CS/corn starch films were similar to the increase with the amount of cinnamaldehyde added, and the changes were from a high content to a low content. When the content of cinnamaldehyde reached 6.4%, the TS value of the film decreased to the lowest value, 4.79 ± 0.19 MPa, which was only 18.24% of that without cinnamaldehyde. The results showed that the addition of cinnamaldehyde had a significant effect on the TS value of the film. At the same time, with the addition of cinnamaldehyde, the E value of the film showed a downward trend as a whole. Through the previous orthogonal experiments, it can be verified that the main factor affecting the tensile properties of the film was glycerin content, and the optimum proportion obtained by orthogonal tests was a comprehensive TS value, and the combination of E value and WVTR value just coincided with the low glycerin content and E value. Therefore, after adding the hydrophobic cinnamaldehyde, the cinnamaldehyde molecules destroyed or affected the arrangement of CS and corn starch molecules, and counteracted the number of hydrogen bonds in the mixed solution. It would lead to a decrease in TS and E values [27]. At the same time, due to the increase in the content of cinnamaldehyde, the local distribution of cinnamaldehyde in the film was uneven, some cinnamaldehyde was cross-linked with CS and corn starch molecules, and the cracking extension of the film was hindered, resulting in the arrangement and slippage of the cinnamaldehyde in the crack region, consuming additional energy. It eventually led to the partial accumulation of cinnamaldehyde, resulting in stress concentrations, which made the E value of the film fluctuate. The E value of 3.2% cinnamaldehyde was 0.95% of the lowest value, and the TS value of 6.4% cinnamaldehyde was only 4.79 ± 0.19 MPa. Due to the high content of cinnamaldehyde, the film forming rate was not high and the performance was very poor. Therefore, the two films with different cinnamaldehyde concentrations were denied.

### 3.4. Water-Vapor Transmission Rate of the CS/Corn Starch/Cinnamaldehyde Films

Table 5 shows the water-vapor transmittance rate of the different cinnamaldehyde ratios of films. The water-vapor transmittance rate of the film decreased obviously as the cinnamaldehyde ratio increased, and the minimum value was 4.91 ± 0.22 ^b^ × 10^−3^ g/m^2^∙s when the cinnamaldehyde ratio was 6.4%; however, thickness of films with different ratios of cinnamaldehyde barely changed. When the cinnamaldehyde ratio was more than 1.6%, WVTR was maintained at a lower level. This is mainly because of the increase of cinnamaldehyde. Cinnamaldehyde is a hydrophobic component, which may change the original molecular structure of CS/corn starch films and reduce the penetration of water molecules [28].

### 3.5. A Study on the Bacteriostatic Effects of the CS/Corn Starch/Cinnamaldehyde Films

The bacteriostatic zone of CS/corn starch/cinnamaldehyde (0%) film can be seen from Figure 4. that PP film had no bacteriostatic effect on all three bacteria, and the bacteriostatic zone was 0 mm. This showed that the CS content played an important role in the antimicrobial performance of the film, which was consistent with the experimental results of Liu et al. [29]. Higher CS contents may cause more proteins to bind to negatively charged lipopolysaccharides, which are easily absorbed by the cell surface, thus inhibiting nutrient transport into cells and ultimately resulting in cell death [30]. 

A CS/corn starch film without cinnamaldehyde had different bacteriostatic effects on three kinds of bacteria, among which the bacteriostatic zones against *Botrytis cinerea* and *Rhizopus* were 12.96 ± 1.13 mm and 11.99 ± 0.24 mm, and against *Escherichia coli* were 3.45 ± 0.87 mm. A large number of researches have proved that chitosan CS itself had a broad-spectrum bacteriostatic effect and had a strong inhibitory effect on a fungus and bacteria [31,32,33]. After adding cinnamaldehyde to the film, the bacteriostatic effect on all three bacteria was improved [34]. In addition, with the increase of cinnamaldehyde content, the bacteriostatic zone value increased, and the bacteriostatic effect on the three bacteria was significant (*p* < 0.05). The bacteriostatic zone of 6.4% cinnamaldehyde content against *Escherichia coli* reached 43.81 ± 0.27 mm. Therefore, the CS/corn starch/cinnamaldehyde film had significant bacteriostatic properties against *Botrytis cinerea* and *Rhizopus* [35,36], which were dominated by the dominant putrefying bacteria of strawberries, and *Escherichia coli*, which is one of the food-borne pathogens. In particular, the added value of cinnamaldehyde at 1.6% or higher had an obvious bacteriostatic effect. 

### 3.6. Study on the Fresh-Keeping Effects of CS/Corn Starch/Cinnamaldehyde Films on Strawberries

According to the color difference and bacteriostatic effects, the color difference of 2.5% was 254.25% of that of 1.6%. When the added cinnamaldehyde exceeded 1.6%, the films began to show better bacteriostasis. From the perspective of strawberry fresh-keeping and aesthetics, a film with a cinnamaldehyde content of 1.6% was selected for preservation experiments.

Figure 5a–f showed weight loss rate, titratable acidity, the ascorbic acid (Vc) content, total soluble solids (TSS), firmness, wrapped rot, and the visual appearance of strawberries after wrapping the strawberries using the films. Figure 5a showed an increasing trend in the weight loss of all samples 13 days after first being stored. The mass-loss rate of each group also increased over time, which was due to the facts that the strawberries were rich in water, and that the breathing and transpiration resulted in increased water losses. After the losses of large amounts of water, strawberries would soften locally, which would aggravate the hydrolysis of enzymes and accelerate the senescence of strawberry cells, reducing their value at the same time. The effect was particularly noticeable on the 7th day. It was because, after the 7th day, some strawberries were seriously infected by fungi, especially in the CK group. On the 7th day, a large number of strawberries began to spoil. Furthermore, the weight loss rate of strawberries in the CK group was significantly higher than those of strawberries wrapped in the film. The main reason was that the moisture in unwrapped strawberries easily escaped into the air [37]. The moisture in the wrapped strawberries was protected by the film, which acted as a barrier to oxygen, carbon dioxide, and moisture. It prevented the water in the strawberries from evaporating to a certain extent, reducing breathing, water loss, and oxidation [38]. The water loss rate of the experimental group was significantly lower than that of the control group. On the 13th day, the weight loss of the CK group was 14.85% ± 0.87%, while that of the cinnamaldehyde film group was only 10.39% ± 1.07%. The water retention performance of the experimental group was better than that of the CK group. It was because cinnamaldehyde had a strong hydrophobicity, which made the water-vapor produced by fruit respiration condense on the film and the fruit surface. It prevented the dissipation of water from the strawberries, reducing the mass-loss rate [39]. It can be seen from the diagram that the water retention of PP group was stronger than that of no cinnamaldehyde film, which is due to the lower permeability coefficient of PP film compared with no cinnamaldehyde film, which can prevent strawberry water evaporation more effectively [40]. 

The titratable acid content represented the sugar-acid ratio in strawberries, which was an important index for mouth-feel. Changes in acidity were significantly affected by the rate of metabolism, especially respiration. Figure 5b showed the titratable acid (TSS) of strawberries in 13 days; the acidity of strawberries first increased and then decreased, which was due to the fact that the maturity of picked strawberries was 7 days or 8 days post-ripening. With the complete maturity of strawberries, the titratable acid value decreased as a whole, and the rate of descent in the initial period of storage was much higher than that in the later period. The downward trend of the CK group was the most obvious. Compared with the control group, the rate of the acidity of strawberry in the experimental group was slower than that of the control group, especially the acidity of strawberry in the cinnamaldehyde film preservation group was significantly higher than that in the CK group. On the 13th day, the acidity value was 0.59% ± 0.05%, which was 141.89% of that in the CK group, which shows an effectively delay of the titratable acid decline rate of strawberries. That was because the packaging can modify the internal atmosphere around the strawberries, and the film had a certain air permeability to form a high carbon dioxide environment, which lowered the respiration intensity of the strawberries during storage, reducing the consumption of titratable acid as a respiration substrate, and therefore, reducing the rate of its decline [41]. 

Figure 5c showed the Vc of strawberries after 13 days of a preservation experiment. The Vc content of the CK group and experimental group decreased overall, as the storage time of strawberries was prolonged. It was due to external oxidation, self-breathing, and microbial infestation during strawberry storage. It could be clearly seen from the figure that all packaging materials inhibited the loss of ascorbic acid in the packaged fruit. In the cinnamaldehyde film group, the Vc content was 48.39 ± 2.78 mg/100 g on the 13th day, while the CK group was only 35.22 ± 1.49 mg/100 g; the former was significantly slower than the latter in terms of Vc maintainance. At the same time, the PP film was slower than the CK group because the PP film can block part of the oxygen to avoid strawberry Vc consumption. The film containing cinnamaldehyde was more effective than PP film at reducing Vc consumption because the former can not only isolate oxygen and microbial infection, but also had certain antibacterial abilities [41]. 

In addition, the content of the soluble solids of a strawberry are an important index to reflect the sugar content of the strawberry’s fruit, and it is also an important part of the evaluation of the nutritional value of strawberry fruit [42]. Figure 5d showed the effects of different treatment groups on the soluble solids of strawberries. As can be seen from this figure, the soluble solids content of strawberries, as a whole, decreased over time, which was due to the fact that strawberries had to decompose their own nutrients to maintain respiration during storage. Soluble solids were the main objects consumed in the process of respiration. In CK group, the soluble solids of strawberries without any treatment decreased most significantly, which was only 4.2 ± 0.29 on the 13th day and 10.5 ± 0.50 on the 1st day, and the downward trend was significant (*p* < 0.05). The value of soluble solids in PP film treatment group was slightly higher than that in CK group, which was due to the fact that PP film could isolate strawberries from infection and corruption by external microorganisms [43]. Moreover, the content of soluble solids in the cinnamaldehyde film and the non-added cinnamaldehyde film in the experimental group were the top two. The content of soluble solids in the cinnamaldehyde film reached 8.4 ± 0.26 at 13th day, which was 200.51% the content of CK group. The results showed that the whole curve of cinnamaldehyde film decreased more slowly than that of CK group, which indicated that the consumption of soluble solids of strawberries in the cinnamaldehyde film group was slower than that in CK group.

Loss of texture is one of the main factors that affect the quality and the post-harvest shelf lives of fruit and vegetables. Therefore, the texture is an important strawberry quality parameter. In addition, the firmness of strawberries during post-harvest storage is generally affected by physicochemical changes due to ripening that continues even after harvest, leading to softening of the fruit. These changes are mainly attributed to the action of pectolytic enzymes on the solubilization of pectin and other cell wall components [44]. During ripening, strawberries soften considerably due to softness of the middle lamella of the cell wall. Cortical parenchyma cells, cell-wall strength, cell-to-cell contact, and cellular turgor can also influence firmness. Figure 5e showed the firmness of strawberry after 13 days of the preservation experiment, and the firmness of strawberries in the control and experimental groups both gradually decreased over time. It was because postharvest strawberries still had high post-harvest physiological activity, and their firm cell walls, cell transfer substances, and swelling pressures were affected by their own respiration [45]. Because the strawberry was not fully mature at the time of picking, the firmness value was higher than the initial value on the 3rd day, and then began to decrease. The obvious downward trend of the experimental group was more delayed than that of the control group. It was because the experimental group’s packaging materials could reduce water loss to a certain extent to avoid strawberry oxidation and reduce respiration, reducing the chance of fungal infection. It also has been reported that CS and other biopolymers are selective O_2_ and CO_2_ barriers thus, can modify the internal atmosphere and slow down the respiration rates of fresh fruits and vegetables [46]. It is worth mentioning that in the experimental group, the firmness of strawberries without the added cinnamaldehyde film was high, while the firmness of cinnamaldehyde film was very low, which may be due to the volatilization of cinnamaldehyde in the cinnamaldehyde film attached to the surface of strawberries or entering the micropore gap of strawberries. Under “infiltration,” although cinnamaldehyde can effectively inhibit the dominant spoilage bacteria of strawberries and prevent them from being infected by microorganisms, it slowed down the respiration rate. It also reduced the firmness of strawberries. Furthermore, strawberries sealed with PP had the best firmness at 13 days, which was considered to be the main reason for the qualitative decrease of fruit acidity [47]. 

Strawberries are highly perishable fruits with high postharvest physiological activities that limit their shelf lives. Generally, the infected areas in such fruit increase gradually with storage time. According to Table 6, strawberry fruits softened and rotted gradually during storage. The CK group had the fastest decay rate, and the decay level reached IV on the 9th day. The table shows that the packaging slowed down the rate of strawberry decay to a certain extent. The decay rate of PP film group was not obvious, as the rotting area was close to 50% on the 7th day, and it reached the highest decay grade on the 13th day. The decay grades of cinnamaldehyde group and the experimental group 231 on the 13th day were III and IV, respectively, which are equivalent to the decayed areas of the CK group on the 7th and 9th days. Those results are because the poor moisture penetrability of the PP film prevented the escape of the moisture produced by respiration, causing it to gather on the surface of the fruit. Moreover, its poor gas barrier property allowed the oxygen in the air to penetrate the PP film, promoting the growth and reproduction of microorganisms on the surface of the fruit, which resulted in rapid decay. The CS contained in the experimental group 231 had certain antibacterial properties and could reduce the microbial activity overall. In addition, the CS in the film provided great water-vapor permeability and mechanical properties, so that the strawberry did not have large areas of decay and maintained a relatively great color and appearance, while the cinnamaldehyde group gave full play to the bacteriostatic performance of cinnamaldehyde on this basis, so the decay rate of cinnamaldehyde group was lower. Thus, the active food packaging films can approximately extend the shelf life of strawberries to 11 days.

## 4. Conclusions

The casting method was used to successfully fabricate CS/corn starch films, and the effects of adding different ratios of cinnamaldehyde on the material properties of CS/corn starch films were studied through a series of tests. The results showed that the water-vapor transmittance rate of the films decreased as the cinnamaldehyde ratio increased, and the minimum value was 4.91 ± 0.22 ^b^ × 10^−3^ g/m^2^∙s when the cinnamaldehyde ratio was 6.4%. The added cinnamaldehyde interacted with CS, making the antibacterial properties of the film better. The preservation experiments showed that the cinnamaldehyde group could inhibit the decline of strawberry freshness and effectively maintain the nutritional value of strawberries. The results showed that the Vc content was 48.39 ± 2.78 mg/100 g in the cinnamaldehyde film group on the 13th day, while in the CK group it was only 35.22 ± 1.49 mg/100 g; the former was significantly slower than the latter Vc. Thus, CS/corn starch/cinnamaldehyde films have great potential in fruit and vegetable preservation as an active food packaging.

## Figures and Tables

**Figure 1 foods-08-00423-f001:**
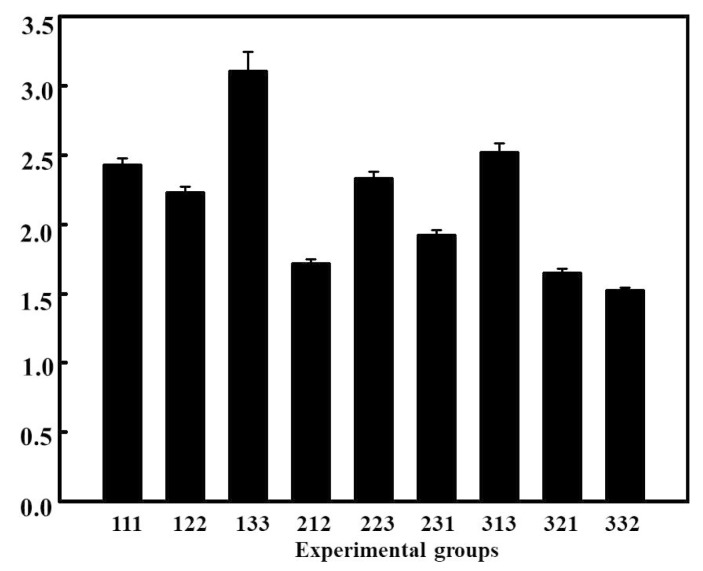
Water-vapor transmission rate of different experimental groups (*p* < 0.05).

**Figure 2 foods-08-00423-f002:**
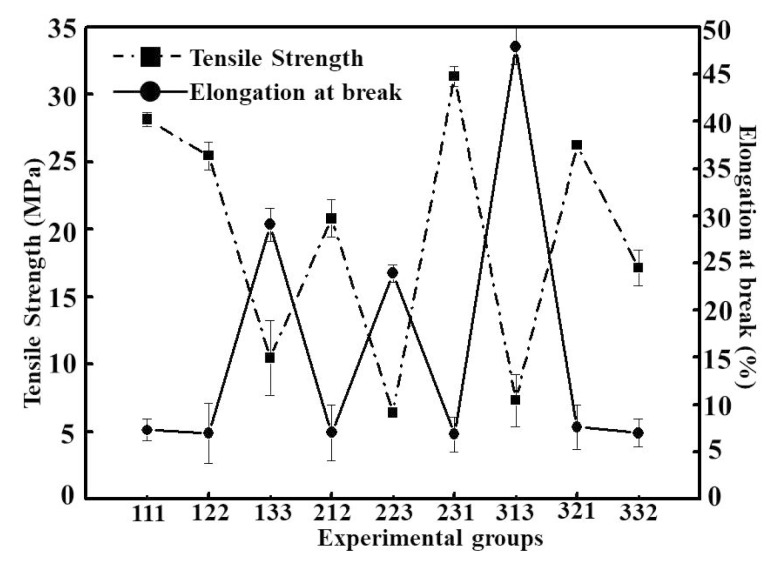
Mechanical properties of different experimental groups (*p* < 0.05).

**Figure 3 foods-08-00423-f003:**
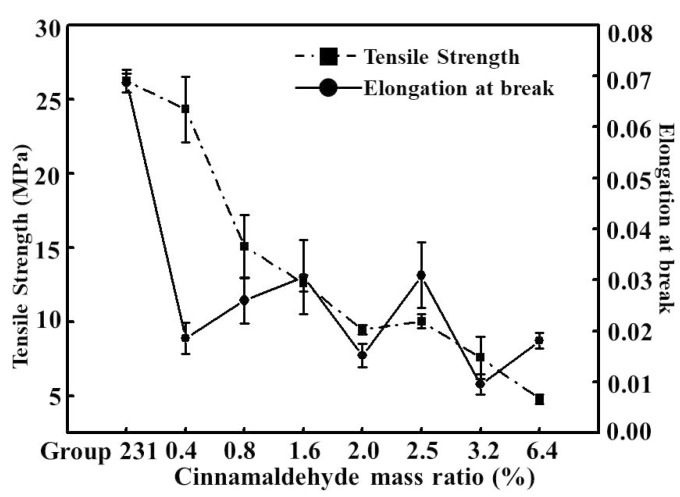
Mechanical properties of CS/corn starch films with different cinnamaldehyde ratios (*p* < 0.05).

**Figure 4 foods-08-00423-f004:**
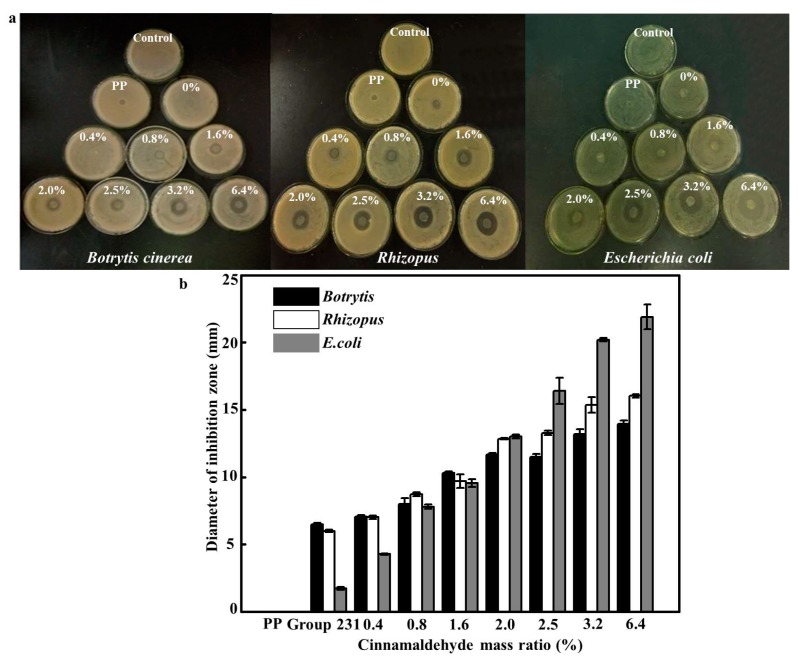
The bacteriostatic effects of CS/corn starch/cinnamaldehyde films (*p* < 0.05).

**Figure 5 foods-08-00423-f005:**
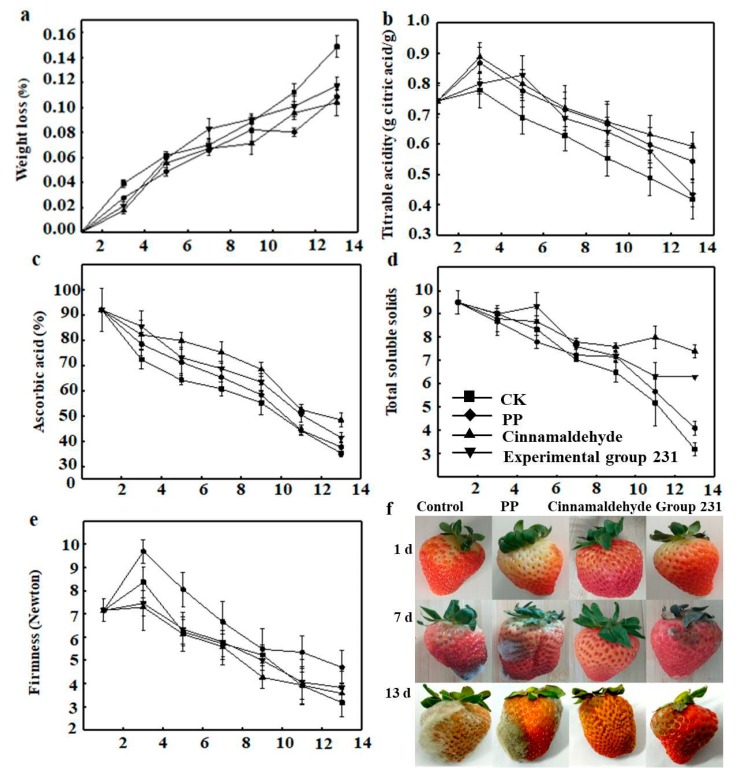
The effects of different films on (**a**) weight loss, (**b**) titratable acidity, (**c**) ascorbic acid content, (**d**) total soluble solids, (**e**) firmness of strawberries stored at room temperature, and (**f**) the visual appearance of packaged strawberries after 13 days storage at room temperature (*p* < 0.05).

**Table 1 foods-08-00423-t001:** L9 (33) orthogonal experiment with different factors and levels.

Level	A: CS (W%)	B: Corn Starch (W%)	C: Glycerin (W%)
1	2%	5%	0.5%
2	2.5%	6%	1%
3	3%	7%	1.5%

**Table 2 foods-08-00423-t002:** The structure of the experimental design.

Experiment Number	A: CS (W%)	B: Corn Starch (W%)	C: Glycerin (W%)	Experimental Result Score
1	1	1	1	61
2	1	2	2	54
3	1	3	3	47
4	2	1	2	65
5	2	2	3	50
6	2	3	1	79
7	3	1	3	51
8	3	2	1	66
9	3	3	2	67
K_1_	54	59	68.67	
K_2_	64.67	56.67	62	
K_3_	61.33	64.33	49.33	
Extreme difference	10.67	7.66	19.34	
Major factor	C > A > B			
Optimal condition	A_2_B_3_C_1_			

**Table 3 foods-08-00423-t003:** Analysis of variance of the physical and chemical properties of the films.

Source	Sum of Squares	Degree of Freedom	Mean Square	*F* Value	*p* Value
Correction model	850.00	6	141.67	35.42	<0.05
Intercept	32,400.00	1	32,400.00	8100.00	<0.01
A	178.67	2	89.33	22.33	<0.05
B	92.67	2	46.33	11.58	>0.05
C	578.67	2	289.33	72.33	<0.05
Error	8.00	2	4.00		
Total	33,258.00	9			
Total corrected	858.00	8			

*R*^2^ = 0.98; adjusted *R*^2^ = 0.977.

**Table 4 foods-08-00423-t004:** Color values of the chitosan (CS)/corn starch/cinnamaldehyde films with different cinnamaldehyde ratios.

Cinnamaldehyde Ratios	L*	a*	b*	ΔE
Standard	72.98 ± 1.15 ^a^	−1.47 ± 0.48 ^a^	14.03 ± 0.64 ^a^	
Experimental group 231	83.91 ± 1.25 ^b^	−0.97±0.56 ^b^	12.68 ± 0.34 ^b^	0.39 ± 0.061 ^a^
0.4%	78.39 ± 0.89 ^c^	−1.78 ± 0.61 ^c^	14.57 ± 0.55 ^c^	1.173 ± 0.086 ^b^
0.8%	78.41 ± 0.46 ^c^	−1.85 ± 0.72 ^c^	14.72 ± 0.56 ^c^	1.294 ± 0.094 ^b^
1.6%	78.49 ± 0.74 ^c^	−1.95 ± 0.52 ^c^	15.24 ± 0.39 ^c^	1.677 ± 0.081 ^b^
2.0%	77.97 ± 0.76 ^c^	−2.19 ± 0.45 ^c^	16.64 ± 0.51 ^d^	2.925 ± 0.111 ^c^
2.5%	76.96 ± 0.81 ^c^	−2.65 ± 0.53 ^d^	17.22 ± 0.50 ^d^	4.175 ± 0.084 ^d^
3.2%	77.13 ± 0.69 ^c^	−2.79 ± 0.45 ^d^	17.55 ± 0.64 ^d^	4.479 ± 0.081 ^e^
6.4%	76.88 ± 0.96 ^c^	−2.86 ± 0.81 ^d^	18.12 ± 0.47 ^e^	5.113 ± 0.092 ^f^

Values are given as: mean ± standard deviation. Different letters in the same column indicate significant differences (*p* < 0.05).

**Table 5 foods-08-00423-t005:** WVTR of different CS/corn starch/cinnamaldehyde films.

Cinnamaldehyde Ratios	Thickness (mm)	WVTR (×10^−3^ g/m^2^∙s)
Experimental group 231	0.112 ± 0.004 ^b^	4.30 ± 0.36 ^c^
0.4%	0.114 ± 0.002 ^a^	4.02 ± 0.16 ^a^
0.8%	0.122 ± 0.001 ^a^	3.14 ± 0.43 ^d^
1.6%	0.125 ± 0.006 ^b^	2.51 ± 0.22 ^b^
2.0%	0.128 ± 0.009 ^c^	2.47 ± 0.12 ^a^
2.5%	0.133 ± 0.002 ^a^	2.35 ± 0.15 ^a^
3.2%	0.134 ± 0.001 ^a^	2.09 ± 0.23 ^b^
6.4%	0.149 ± 0.006 ^b^	1.99 ± 0.15 ^a^

Notes: Thickness of each film was measured by a thickness gauge; WVTR represents water-vapor transmittance rate.

**Table 6 foods-08-00423-t006:** Wrapped rot of packaged strawberries after 13 days of storage at room temperature.

Processing Method/Number of Days	Day 1	Day 3	Day 5	Day 7	Day 9	Day 11	Day 13
CK	Ⅰ	Ⅱ	Ⅱ	Ⅲ	Ⅳ	Ⅳ	Ⅴ
PP	Ⅰ	Ⅱ	Ⅱ	Ⅲ	Ⅲ	Ⅳ	Ⅴ
Cinnamaldehyde	Ⅰ	Ⅰ	Ⅰ	Ⅰ	Ⅱ	Ⅱ	Ⅲ
Experimental group 231	Ⅰ	Ⅰ	Ⅰ	Ⅱ	Ⅲ	Ⅲ	Ⅳ

Notes: Grade I, no rot; Grade II, there are 1–3 small decay spots on the fruit surface; Grade Ⅲ, the decayed area accounts for 25% to 50% of the fruit area; Grade Ⅳ, the decayed area accounts for 75% to 100% of the fruit area.

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
