# Peer review of "Preparation of Chitosan/Corn Starch/Cinnamaldehyde Films for Strawberry Preservation"

_foods, 2019, doi:10.3390/foods8090423_

Round 1

Reviewer 1 Report

The object of the manuscript is utmost actual; developing active, shelf-life extending packaging materials has a high relevance. Although the presented research topic and the results have scientific value, the manuscript needs such improvements that it can not be recommended to be published in Foods at this stage.

Thorough and profound revision is needed to improve the quality of the paper.

Comments and suggestions are listed below.

The prepared chitosan/corn starch/cinnamaldehyde film should not be referred as a composite - in composites the components remain separate and distinct in the final structure which is not the case here since the starch was gelatinized. The prepared material is much more a blend. The aim was to prepare films for active food packaging applications – it is recommended to us the term ”active packaging” LN 60-61 Cinnamaldehyde containing active films have been widely studied in the literature. The published studies should be explored more.

For example, see:

Ouattara, B., Simard, R. E., Piette, G., Bégin, A., & Holley, R. A. (2000). Inhibition of surface spoilage bacteria in processed meats by application of antimicrobial films prepared with chitosan. International journal of food microbiology62(1-2), 139-148.

Balaguer, M. P., Lopez-Carballo, G., Catala, R., Gavara, R., & Hernandez-Munoz, P. (2013). Antifungal properties of gliadin films incorporating cinnamaldehyde and application in active food packaging of bread and cheese spread foodstuffs. International journal of food microbiology166(3), 369-377.

de Souza, A. C., Dias, A. M., Sousa, H. C., & Tadini, C. C. (2014). Impregnation of cinnamaldehyde into cassava starch biocomposite films using supercritical fluid technology for the development of food active packaging. Carbohydrate polymers102, 830-837.

Higueras, L., López-Carballo, G., Gavara, R., & Hernández-Muñoz, P. (2015). Reversible covalent immobilization of cinnamaldehyde on chitosan films via schiff base formation and their application in active food packaging. Food and bioprocess technology8(3), 526-538.

In general, scientific language should be improved and the manuscript needs proofreading to eliminate the grammatical and phrasing (using inappropriate idioms) errors. To name but a few:

      LN 61 “anti-bacteria solution”

LN 94 “Ccorn”

LN 99 “, add CS then stir”

LN 150 “triplicates”

LN 405 “vacuum ultrasonic”

LN 16-17 “And the composite film showed the composite film had”

LN 289 – Table 4. “Bath”

LN 291 “indicate significantly different” correct to significant difference

LN 307,310,314 “Cinnamaldehydes” with an “s” can be read

LN 42 “kind of green”

LN 60-61 needs complete rephrasing

LN 65 “obviously” no need to use

LN 66-67 needs complete rephrasing

LN 83 “oil of CS”

Ln 106-108 needs complete rephrasing

LN 123 “anti-bubble …. film”

LN 173 what is “normal temperature”

LN 188 “floating platform”

LN 190 “firm measurement” correct to firmness

LN 197 “NaOH spent for titration”

LN 199 “percent citric acid” use citric acid equivalent

LN 202-203 needs complete rephrasing

LN 221 “linear corn starch” high amylose corn starch is more appropriate (through the manuscript)

LN 221 “film is densely packed” “tidy”

LN 234 “had been famous”

LN 252 “rot” should not be applied to express biodegradation or deterioration (LN 354)

The “degradation” should not be applied to express the degraded material.

LN 278 “brightness” should be corrected to lightness

LN 264-268 needs complete rephrasing

LN 340 “beauty”

LN 445 “PP film group was not obvious”

LN 468-469 needs rephrasing

There are two “2.2” By definition degradation and biodegradation are not exactly the same. Please consider this difference in the manuscript. LN 15-16 clarification is needed – Which performance of the film was excellent when the elongation at break was 6.87% etc.? (Also, I would recommend avoiding the usage of “excellent”) LN 110 -111 What is meant by 40 and 30 minutes? And what were the parameters for the mentioned tests. The same as in 2.3 and 2.6? LN 152-154 should be replaced after 2.2 (second) LN 188-189 as one can guess the method described is about to determine flash firmness. Please name and show the measurement correctly (with equipment type) LN 268 “Many researchers”, whilst the number of references is only two – please add more Figure 3 caption should be corrected (“diagram”) LN 273 Use simply the before mentioned “Film color difference” LN 276-277 L* decreased and b* increased “successively” – was that a goal? LN 285-288 “Changes in color….” as stated before the cinnamaldehyde has a yellowish green color – changes in color characteristic could be related to the color of the additive. The degradation test was performed on films without cinnamaldehyde – what was the reason for that? Couldn’t the cinnamaldehyde affect the degradation? The test should be conducted on the final film/films. Also, test parameters should be added - at least environmental (e.g. temperature during the experiment )  In Table 4 significant difference is presented, however, nowhere else LN 323 Were polymer composite fibers forming on the cell surface? Was the case here also? Or it is just a referee to an observation of different authors? LN 318-320 Besides rephrasing is needed, how could PP results show that CS content was directly related to the antimicrobial performance of the film? LN 338-339 How is it related to the antibacterial effect? LN 353 What is CK group? 8. a, b, c, d, e – quality of figures should be improved It would be necessary to clarify the test method used for fruit deterioration test. It is not clear how the films could be act as barriers against air and moisture. How was the package sealed? If it was not sealed or not properly (airtight), then they could not act as gas or moisture barriers. Gas permeability or at least water vapor permeability test (which can be performed more easily) should be conducted to determine the barrier properties. To make conclusions regarding the barrier properties these kinds of test should be performed. LN 363-365 It is correct that cinnamaldehyde is a hydrophobic component in the film, however, since the change in hydrophilicity of the film were not tested, we do not know anything about the hydrophobicity of the films. (the tested film contained only 1.6% cinnamaldehyde). LN 369-370 Please clarify what kind of nanocomposite was this, and how is it related to the current research.

Author Response

Comments and Suggestions for Authors 1:

The object of the manuscript is utmost actual; developing active, shelf-life extending packaging materials has a high relevance. Although the presented research topic and the results have scientific value, the manuscript needs such improvements that it can not be recommended to be published in Foods at this stage. Thorough and profound revision is needed to improve the quality of the paper. Comments and suggestions are listed below.

The prepared chitosan/corn starch/cinnamaldehyde film should not be referred as a composite - in composites the components remain separate and distinct in the final structure which is not the case here since the starch was gelatinized. The prepared material is much more a blend. The aim was to prepare films for active food packaging applications – it is recommended to us the term ”active packaging” LN 60-61 Cinnamaldehyde containing active films have been widely studied in the literature. The published studies should be explored more.

For example, see:

Ouattara, B., Simard, R. E., Piette, G., Bégin, A., & Holley, R. A. (2000). Inhibition of surface spoilage bacteria in processed meats by application of antimicrobial films prepared with chitosan. International journal of food microbiology, 62(1-2), 139-148.

Balaguer, M. P., Lopez-Carballo, G., Catala, R., Gavara, R., & Hernandez-Munoz, P. (2013). Antifungal properties of gliadin films incorporating cinnamaldehyde and application in active food packaging of bread and cheese spread foodstuffs. International journal of food microbiology, 166(3), 369-377.

de Souza, A. C., Dias, A. M., Sousa, H. C., & Tadini, C. C. (2014). Impregnation of cinnamaldehyde into cassava starch biocomposite films using supercritical fluid technology for the development of food active packaging. Carbohydrate polymers, 102, 830-837.

Higueras, L., López-Carballo, G., Gavara, R., & Hernández-Muñoz, P. (2015). Reversible covalent immobilization of cinnamaldehyde on chitosan films via schiff base formation and their application in active food packaging. Food and bioprocess technology, 8(3), 526-538.

Re: Corrected and added. Thanks.

In general, scientific language should be improved and the manuscript needs proofreading to eliminate the grammatical and phrasing (using inappropriate idioms) errors. To name but a few:

Re: Corrected. Thanks.

LN 61 “anti-bacteria solution”

Re: Corrected. Thanks.

LN 94 “Ccorn”

Re: Corrected. Thanks.

LN 99 “, add CS then stir”

Re: Corrected. Thanks.

LN 150 “triplicates”

Re: Corrected. Thanks.

LN 405 “vacuum ultrasonic”

Re: Corrected. Thanks.

LN 16-17 “And the composite film showed the composite film had”

Re: Corrected. Thanks.

LN 289 – Table 4. “Bath”

Re: We used “experimental group” instead of “Bath”. Thanks.

LN 291 “indicate significantly different” correct to significant difference

Re: Done. Thanks.

LN 307,310,314 “Cinnamaldehydes” with an “s” can be read

Re: Revised. Thanks.

LN 42 “kind of green”

Re: Corrected. Thanks.

LN 60-61 needs complete rephrasing

Re: We corrected the wrong description. Thanks.

LN 65 “obviously” no need to use

Re: Corrected. Thanks.

LN 66-67 needs complete rephrasing

Re: Done. Thanks.

LN 83 “oil of CS”

Re: Corrected. Thanks.

Ln 106-108 needs complete rephrasing

Re: Done. Thanks.

LN 123 “anti-bubble …. film”

Re: Corrected. Thanks.

LN 173 what is “normal temperature”

Re: Corrected. Thanks.

LN 188 “floating platform”

Re: Corrected. Thanks.

LN 190 “firm measurement” correct to firmness

Re: Corrected. Thanks.

LN 197 “NaOH spent for titration”

Re: We corrected the wrong description. Thanks.

LN 199 “percent citric acid” use citric acid equivalent

Re: Done. Thanks.

LN 202-203 needs complete rephrasing

Re: We rewrote this part. Thanks.

LN 221 “linear corn starch” high amylose corn starch is more appropriate (through the manuscript)

Re: Corrected. Thanks.

LN 221 “film is densely packed” “tidy”

Re: Corrected. Thanks.

LN 234 “had been famous”

Re: Corrected. Thanks.

LN 252 “rot” should not be applied to express biodegradation or deterioration (LN 354)

Re: Revised. Thanks.

The “degradation” should not be applied to express the degraded material.

Re: We used “degradable property” instead. Thanks.

LN 278 “brightness” should be corrected to lightness

Re: Corrected. Thanks.

LN 264-268 needs complete rephrasing

Re: Done. Thanks.

LN 340 “beauty”

Re: Corrected. Thanks.

LN 445 “PP film group was not obvious”

Re: Corrected. Thanks.

LN 468-469 needs rephrasing

Re: Done. Thanks.

There are two “2.2” By definition degradation and biodegradation are not exactly the same. Please consider this difference in the manuscript.

Re: Degradable materials include materials that can be degraded by physical and biological factors, while biodegradation refers only to materials that can be completely digested as food for microorganisms. The role of microorganisms in degradation has not been considered separately in this experiment. Therefore, we chose the term degradation.[Condecid, M.; Fernandezcalvino, D.; Novoamunoz, J. C.; Ariasestevez, M.; Diazravina, M. Degradation of sulfadiazine, sulfachloropyridazine and sulfamethazine in aqueous media, J. Environ. Manage. 2018, 228, 239-248.][ Shukla, A. K.; Upadhyay, S. N.; Dubey, S. K. Current trends in trichloroethylene biodegradation: a review. Crit. Rev. Biotechnol. 2014, 34, 101-114.][ Cheng, X.; Hou, D.; Xu, C. The effect of biodegradation on adamantanes in reservoired crude oils from the Bohai Bay Basin, China. Organic Geochemistry, Org. Geochem. 2018, 123, 38-43.]. Thanks.

LN 15-16 clarification is needed – Which performance of the film was excellent when the elongation at break was 6.87% etc.? (Also, I would recommend avoiding the usage of “excellent”)

Re: Yes, Done. Thanks.

LN 110 -111 What is meant by 40 and 30 minutes? And what were the parameters for the mentioned tests. The same as in 2.3 and 2.6?

Re: We changed “minute” into “score”. The total score is set to 100, in which the water permeability, tensile strength and elongation at break account for 4:3:3, respectively. Because our composite film is mainly used for preservation, the barrier property is more important than other properties, and the parameters for the mentioned tests are the same as 2.3 and 2.6. Thanks.

LN 152-154 should be replaced after 2.2 (second)

Re: Done. Thanks.

LN 188-189 as one can guess the method described is about to determine flash firmness. Please name and show the measurement correctly (with equipment type)

Re: Added. Thanks.

LN 268 “Many researchers”, whilst the number of references is only two – please add more Figure 3 caption should be corrected (“diagram”)

Re: Added and corrected. Thanks.

LN 273 Use simply the before mentioned “Film color difference”

Re: Corrected. Thanks.

LN 276-277 L* decreased and b* increased “successively” – was that a goal?

Re: No, that might be a narrative, so I deleted it. Thanks.

LN 285-288 “Changes in color….” as stated before the cinnamaldehyde has a yellowish green color – changes in color characteristic could be related to the color of the additive. The degradation test was performed on films without cinnamaldehyde – what was the reason for that? Couldn’t the cinnamaldehyde affect the degradation?

Re: We apologized that the name of the composite film was incorrect, cinnamaldehyde was missed, like Table 4 caption. When we designed the experiment, the tests were all carried out on a composite film containing cinnamaldehyde after the orthogonal experiment. Due to the degradable of cinnamaldehyde, cinnamaldehyde does not affect the degradation performance of the composite films [Kenawy, E.; Omer, A. M.; Tamer, T. M. Elmeligy, M. A.; Mohy Eldin, M. S. Fabrication of biodegradable gelatin/chitosan/cinnamaldehyde crosslinked membranes for antibacterial wound dressing applications, Int. J. Biol. Macromol. 2019, 139, 440-448.].Thanks.

The test should be conducted on the final film/films. Also, test parameters should be added - at least environmental (e.g. temperature during the experiment )

Re: Yes, all tests were conducted on the final films, and we added the temperature to 2.4, 2.5 and 2.6. Thanks.

In Table 4 significant difference is presented, however, nowhere else LN 323 Were polymer composite fibers forming on the cell surface? Was the case here also? Or it is just a referee to an observation of different authors?

Re: LN323 were re-written. Thanks.

LN 318-320 Besides rephrasing is needed, how could PP results show that CS content was directly related to the antimicrobial performance of the film?

Re: Yes, some parts were incorrect. We wanted to express that CS content played an important role in the antimicrobial performance, and pure starch films have poor antibacterial properties [ Wikman, J.; Blennow, A.; Buléon, A. Influence of amylopectin structure and degree of phosphorylation on the molecular composition of potato starch lintners, Biopolymers, 2013, 101, 257-271.] [Sukhija, S.; Singh, S.; Riar, C. S. Physicochemical, crystalline, morphological, pasting and thermal properties of modified lotus rhizome (Nelumbo nucifera), starch, Food Hydrocolloids, 2016, 60, 50-58.]. In order to avoid the misunderstanding, we added a “0%” in the first sentence of this paragraph. Because we first compared PP film and the composite film without cinnamaldehyde, and then compared the composite films between containing only cinnamaldehyde and containing only CS in the next paragraph. So that we can get accurately analyze the antibacterial effect of CS and cinnamaldehyde respectively. [ Munhuweyi K., Caleb O. J., Lennox C. L., Reenen A. J., LinusOpara U., In vitro and in vivo antifungal activity of chitosan-essential oils against pomegranate fruit pathogens, Postharvest Biol. Technol. 2017, 129, 9-22 ][ Liu, G. Q.; Donoghue, A. M.; Moyle, J. R.; Reyesherr, I.; Blore, P. J.; Bramwell, R. K.; Yoho, D. E.; Venkitanar, K.; Donoghue, D. J. Effects of trans-cinnamaldehyde on campylobacter and sperm viability in chicken semen after in vitro storage, Inter. J. Poultry Sci. 2012, 11, 536-540.]. Thanks.

LN 338-339 How is it related to the antibacterial effect?

Re: Sorry, We corrected the wrong description. We put them to the first paragraph of the preservation performance. Thanks.

LN 353 What is CK group? 8. a, b, c, d, e – quality of figures should be improved It would be necessary to clarify the test method used for fruit deterioration test. It is not clear how the films could be act as barriers against air and moisture. How was the package sealed? If it was not sealed or not properly (airtight), then they could not act as gas or moisture barriers. Gas permeability or at least water vapor permeability test (which can be performed more easily) should be conducted to determine the barrier properties. To make conclusions regarding the barrier properties these kinds of test should be performed.

Re: I described CK group in 2.9: Samples without film was used as a control group named CK group. I have added this clarification in 2.9. And Water vapor permeability test results were added in 3.5. Every three strawberries are packed in a sealed bag (15 * 15 cm) made of a composite films. All samples were placed on a glass plate and stored at room temperature (20 ± 1 °C) and relative humidity (65% ± 5%). Thanks.

LN 363-365 It is correct that cinnamaldehyde is a hydrophobic component in the film, however, since the change in hydrophilicity of the film were not tested, we do not know anything about the hydrophobicity of the films. (the tested film contained only 1.6% cinnamaldehyde).

Re: Although we did not conduct a hydrophobic test, there are many reports that a small amount of cinnamaldehyde can increase the hydrophobicity of the composite film. [Perdones, Á.; Vargas, M.; Atarés, L.; Chiralt, A. Physical, antioxidant and antimicrobial properties of chitosan–cinnamon leaf oil films as affected by oleic acid, Food Hydrocolloids, 2014, 36, 256-264.][ Chen, H.; Hu, X.; Chen, E.; Wu, S.; Mcclements, D. J. Preparation, characterization, and properties of chitosan films with cinnamaldehyde nanoemulsions, Food Hydrocolloids, 2016, 61, 662-671.][ Otoni, C. G.; Avenabustillos, R. J.; Olsen, C. W.; Bilbaosainz, C.; Mchugh, T. H. Mechanical and water barrier properties of isolated soy protein composite edible films as affected by carvacrol and cinnamaldehyde micro and nanoemulsions, Food Hydrocolloids. 2016, 57, 72-79.] Kenawy et al. investigated the effect of changes in cinnamaldehyde content on film wettability using water contact angle measurements and found that due to the hydrophobicity of cinnamaldehyde, the gelatin/chitosan/cinnamaldehyde film has a high hydrophobicity with a contact angle of 62° in which only 0.5 mL cinnamaldehyde was added.[ Kenawy, E.; Omer, A. M.; Tamer, T. M.; Elmeligy, M. A.; MohyEldin, M. S. Fabrication of biodegradable gelatin/chitosan/cinnamaldehyde crosslinked membranes for antibacterial wound dressing applications, Int. J. Biol. Macromol. 2019, 139, 440-118.].Thanks.

LN 369-370 Please clarify what kind of nanocomposite was this, and how is it related to the current research.

Re: Sorry, it’s a wrong description, we delete it. Thanks.

Reviewer 2 Report

The authors have prepared a paper with certain interest since although other authors have covered chitosan they have combined chitosan with other two compounds.

It would be interesting if the authors described in the introduction studies similar and results found and lacks that will be covered with this research.

It is important to mention the origen of the materials overall chitosan the degree of decetylacion is important but also it is very important to mention the origen (lobter, crab...) since in order to compare with other studies this information is relevant.

It is necessary to modify the latin names all need to be in cursive in the text. The design needs to be justify in the section why the maximum and minims have been selected based in film properties or previous studies please justify.

The statistic need to be included in graphs and tables when is possible, for example figure 1 or fig 8.

The conclusion need to be enhanced in order to give more information regarding the optimum conditions.

Author Response

Comments and Suggestions for Authors 2:

The authors have prepared a paper with certain interest since although other authors have covered chitosan they have combined chitosan with other two compounds.

It would be interesting if the authors described in the introduction studies similar and results found and lacks that will be covered with this research.

Re: Done. Thanks.

It is important to mention the origen of the materials overall chitosan the degree of deacetylacion is important but also it is very important to mention the origen (lobter, crab...) since in order to compare with other studies this information is relevant.

Re: CS with the deacetylation degree of 95% and the molecular weight (Mw) of 2.8 × 105 was purchased from Golden-Shell Pharmaceutical Co. Ltd. (Zhejiang, China), principally from crabs and shrimps. Thanks.

It is necessary to modify the latin names all need to be in cursive in the text. The design needs to be justify in the section why the maximum and minims have been selected based in film properties or previous studies please justify.

Re: Corrected. And the reason why the maximum and minims have been selected based on our previous studies. Thanks.

The statistic need to be included in graphs and tables when is possible, for example figure 1 or fig 8.

Re: Added. Thanks.

The conclusion need to be enhanced in order to give more information regarding the optimum conditions.

Re:Done. Thanks.

Round 2

Reviewer 1 Report

The authors made efforts to improve the quality of the manuscript and to eliminate the scientific language and grammatical error.

e.g.: LN 195 "solvent" -there is nothing to do with solvent here
LN 187-189
LN 324 "degradable property" do the authors mean degraded product/fragmented film pieces?
LN 330 "as the folding became more and more" sounds unscientific
LN 331 "the thin films appeared voids" do the authors mean that voids appeared on/in the films?
LN 542 "the firmness ....was the most obvious"
LN 547 "strawberry coated PP had the best firmness"

LN 591 "obviously" should not be used

LN 593 "the addition of" should be corrected to "the added cinnamaldehyde"
LN 593-595 needs rephrasing

These were just few examples, please go through the manuscript once again.

The degradation seems to be really rapid from Day 29 to Day 33. Isn't there a mistake in the order of the photos? The authors states that the cinnamaldehyde could not affect the degradation of the films - on the other hand they also stated that the additive cause hydrophobic shift. Are the authors certain, that the cinnamaldehyde has no impact on the degradation performance, especially when considering hydrolytic degradation as a starting degradation process? The authors do not have to use every single time the term high amylose corn starch - the recommendation aimed only the term "linear corn starch" Still, it is not clear how the sealing of the films were prepared. The authors used the suggested studies which were only examples. Please do a proper literature review in the field of cinnamaldehye as an active film additive. Fig. 8 - Bath231 should be corrected

Author Response

The authors made efforts to improve the quality of the manuscript and to eliminate the scientific language and grammatical error.

e.g.: LN 195 "solvent" -there is nothing to do with solvent here

Re: Corrected. Thanks.

LN 324 "degradable property" do the authors mean degraded product/fragmented film pieces?

Re: We deleted this part, because the experimental design for degradable property was inappropriate. Thanks.

LN 330 "as the folding became more and more" sounds unscientific

Re: Deleted. Thanks.

LN 331 "the thin films appeared voids" do the authors mean that voids appeared on/in the films?

Re: Corrected. Thanks.

LN 542 "the firmness ....was the most obvious"

Re: Revised. Thanks.

LN 547 "strawberry coated PP had the best firmness"

Re: Corrected. Thanks.

LN 591 "obviously" should not be used

Re: Corrected. Thanks.

LN 593 "the addition of" should be corrected to "the added cinnamaldehyde"

Re: Revised. Thanks.

LN 593-595 needs rephrasing

Re: Done. Thanks.

These were just few examples, please go through the manuscript once again.

Re: Yes, we corrected the wrong description. Thanks.

The degradation seems to be really rapid from Day 29 to Day 33. Isn't there a mistake in the order of the photos? The authors states that the cinnamaldehyde could not affect the degradation of the films - on the other hand they also stated that the additive cause hydrophobic shift. Are the authors certain, that the cinnamaldehyde has no impact on the degradation performance, especially when considering hydrolytic degradation as a starting degradation process? The authors do not have to use every single time the term high amylose corn starch - the recommendation aimed only the term "linear corn starch" Still, it is not clear how the sealing of the films were prepared. The authors used the suggested studies which were only examples. Please do a proper literature review in the field of cinnamaldehye as an active film additive. Fig. 8 - Bath231 should be corrected

Re: We found the experimental design for degradable property was inappropriate, so we decided to remove this part. And, the wrong expression about the starch was corrected. Every three strawberries were placed on acrylic plates and wrapped around with different films (Salgado, P. R , López-Caballero, M. E., Gómez-Guillén, M. C. Sunflower protein films incorporated with clove essential oil have potential application for the preservation of fish patties, Food Hydrocolloids, 2013, 33, 7484.) (Wu, J. L., Ge, S. Y., Liu, H. Properties and antimicrobial activity of silver carp (Hypophthalmichthys molitrix) skin gelatin-chitosan films incorporated with oregano essential oil for fish preservation, Food Packaging and Shelf Life, 2014, 2, 7-16.)(Visvalingam, J., Palaniappan, K., Holley, R. A. In vitro enhancement of antibiotic susceptibility of drug resistant Escherichia coli by cinnamaldehyde, Food Control. 2017, 79, 288-291.). Fig. 8 was replaced. Thanks.